# Enhancing Thermal Stability and Bioaccesibility of Açaí Fruit Polyphenols through Electrohydrodynamic Encapsulation into Zein Electrosprayed Particles

**DOI:** 10.3390/antiox8100464

**Published:** 2019-10-09

**Authors:** Carol López de Dicastillo, Constanza Piña, Luan Garrido, Carla Arancibia, María José Galotto

**Affiliations:** 1Food Packaging Laboratory (Laben-Chile), Department of Science and Food Technology, Faculty of Technology, University of Santiago de Chile, Obispo Umaña 050, 9170201 Santiago, Chile; constanza.pina@usach.cl (C.P.); luan.garrido@usach.cl (L.G.); maria.galotto@usach.cl (M.J.G.); 2Center for the Development of Nanoscience and Nanotechnology (CEDENNA), 9170124 Santiago, Chile; 3Food Properties Research Group, Department of Science and Food Technology, Faculty of Technology, University of Santiago de Chile, Obispo Umaña 050, 9170201 Santiago, Chile; carla.arancibia@usach.cl

**Keywords:** encapsulation, electrospinning, polyphenol, açaí (*Euterpe oleracea* Mart.), zein

## Abstract

The açaí fruit *(Euterpe oleracea* Mart.) is well known for its high content of antioxidant compounds, especially anthocyanins, which provide beneficial health properties. The incorporation of this fruit is limited to food products whose processing does not involve the use of high temperatures due to the low thermal stability of these functional components. The objective of this work was the encapsulation of açaí fruit antioxidants into electrosprayed zein, a heat-resistant protein, to improve their bioavailability and thermal resistance. First, the hydroalcoholic açaí extract was selected due to its high polyphenolic content and antioxidant capacities, and, subsequently, it was successfully encapsulated in electrosprayed zein particles. Scanning electron microscopy studies revealed that the resulting particles presented cavities with an average size of 924 nm. Structural characterization by Fourier transform infrared spectroscopy revealed certain chemical interaction between the active compounds and zein. Encapsulation efficiency was approximately 70%. Results demonstrated the effectiveness of the encapsulated extract on protecting polyphenolic content after high-temperature treatments, such as sterilization (121 °C) and baking (180 °C). Bioaccesibility studies also indicated an increase of polyphenols presence after in vitro digestion stages of encapsulated açaí fruit extract in contrast with the unprotected extract.

## 1. Introduction

In the last decade, the market for nutraceuticals has grown enormously due to accentuated interest by consumers in their therapeutic effects for health disorders, including neurodegenerative and cardiovascular diseases [1,2]. A broad variety of active compounds from natural sources has been researched. The main active compounds from plants are polyphenols, secondary metabolites that make up a large family of substances, from simple molecules to complex structures [3]. Numerous studies have shown that certain fruit contains high levels of antioxidant active compounds. Specifically, one fruit with great antioxidant capacity is açaí (*Euterpe oleracea* Mart.), which has recently emerged as a promising source of energy, and nutritional and antimicrobial properties [4,5]. Preventing oxidative stress in human endothelial cells and the therapeutic effect on neurodegenerative diseases have emerged as bioactivities related to this fruit [6,7,8,9]. Açaí is a palm native to South America that grows mostly in the Amazon estuary, in the north of Brazil. Some studies have also revealed this fruit significantly reduces the risk of atherosclerosis through antioxidant and anti-inflammatory activities [10,11]. The principal flavonoid responsible for this anti-inflammatory activity is the flavone velutin [12]. Açaí fruit is also composed by high content of polyphenolic compounds, especially anthocyanins, as major components cyanidin-3-glucoside and cyanidin-3-rutinoside, and phenolic acids [13,14,15]. Açaí is commonly sold as dehydrated powder to be added to food, in dietary supplements or beverages. However, its current applications are limited to certain foods that do not include such high thermal processes as baking and cooking due to the low stability of the polyphenols, principally the anthocyanins. These molecules can be degraded at increased temperatures, leading to the loss of their functional properties [16]. Recently, a kinetic study of anthocyanin’s açaí thermal degradation by Costa et col. (2018) revealed their degradations fitted a kinetic model of the first order [17]. On the other hand, it is also well known that for natural bioactive compounds to present real benefits, they must be available for absorption after the process of gastrointestinal digestion [18,19]. Thus, encapsulation of natural bioactive compounds is an interesting alternative for performing a double purpose of extending possibilities of incorporation into broader food matrices and enhancing bioavailability [20].

Although several procedures have been used to encapsulate active compounds, such as spray drying, lyophilization, and emulsification, these techniques present disadvantages, such as the complexity of the equipment, use of high temperatures, non-uniform conditions in the drying chamber and lack of particle size control [21]. In recent years, a technology that has received special attention is electrospinning and/or electrospraying [22]. Research studies have clearly shown electrospraying and electrospinning are techniques with functional advantages such as sustained release property, high encapsulation efficiency, and enhanced stability of encapsulated food bioactive compounds [22]. This technique consists of spinning polymeric solutions through high electric fields that exceed the forces of surface tension in the solution of charged polymers. At a certain voltage, fine jets of solution are expelled from the capillary to the collector plate. The solvent evaporates and the segments of fibers or particles are deposited randomly on a substrate. Depending on the specific conditions of polymer solution and the equipment, the process can result in a stretched jet or dispersion of droplets [23]. Several bioactive substances have been successfully encapsulated into electrosprayed particles by using a wide variety of natural polymers as encapsulating materials, depending on the compound to be encapsulated [24]. Among the edible materials, carbohydrates, lipids, and protein have gained the most interest. The latter have numerous advantages, such as increasing the bioavailability of the encapsulated compounds and high binding capacity with active compounds [25]. Corn zein protein has been shown to be a protein resistant to temperatures above 200 °C and has been used as an encapsulation material for some compounds, such as curcumin, improving stability against different values of pH and ultraviolet (UV) radiation [26,27]. Thus, this work presents the selection of a powerful açaí fruit extract, based on its highest phenolic content and antioxidant activities, to be further encapsulated into electrosprayed zein capsules. Although the limited use of açaí fruit in food formulations is evident due to its thermal instability, few works have developed alternatives to protect its active compounds. These encapsulated structures were morphological and structurally characterized and considered a suitable shell to impart thermal protection and enhance the bioavailability of phenolic compounds.

## 2. Materials and Methods

### 2.1. Test Materials and Reagents

Freeze-dried and milled organic açaí fruit was obtained from “Healthy Foods”. Zein (Z 3625), 2,2-diphenyl-1-picrylhydrazyl (DPPH), 2,2′-azinobis(3-ethylbenzothiazoline-6-sulphonate) (ABTS), Folin–Ciocalteu phenol reagent, anhydrous sodium carbonate, gallic acid (GA), ferric 2,4,6-tripyridyl-s-triazine (TPTZ) and 6-hydroxy-2,5,7,8-tetramethylchroman-2-carboxylic acid (Trolox) were obtained from Sigma–Aldrich (Santiago, Chile). Lipase (L3126) and pancreatin (P1750) from porcine pancreas, pepsin (P6887) from porcine gastric mucosa, and porcine bile extract (B8631) were purchased from Sigma–Aldrich (Sigma–Aldrich S.A., USA). NaOH, HCl and different salts to prepare simulated digestion fluids (KCl, KH_2_PO_4_, NaHCO_3_, NaCl, MgCl_2_.(H_2_O)_6_, (NH_4_)_2_CO_3_, and CaCl_2_.(H_2_O)_2_) were purchased from Merck (Merck KGaA S.A., Darmstadt, Germany).

### 2.2. Selection of Açaí Fruit Extract

#### 2.2.1. Preparation of Açaí Fruit Extracts

Active compounds from açaí fruit were extracted using absolute ethanol, ethanol 50%, and distilled water in a 1:300 (solid (g):solvent (mL)) ratio to study the effect of the solvent polarity on the extraction capacity of the most relevant antioxidant compounds. The extractions were carried out at 40 °C for 3 h with an agitation of 150 rpm. Samples were centrifuged, filtered, and used for the antioxidant assays and the analysis of polyphenolic content (PC). Extracts obtained were named “Aç1, Aç2, Aç3” for extracts under ethanol, ethanol 50% and water, respectively. 

#### 2.2.2. Determination of Total Phenolic Content and Antioxidant Activity Studies 

Total phenolic content (TPC) of the extracts was determined following the Folin–Ciocalteu method [28]. 100 μL of each extract was mixed with 3100 μL of distilled water and 200 μL of Folin–Ciocalteu reagent. The samples were taken to darkness for 5 min and 600 μL of anhydrous sodium carbonate at 20% (w/v) was added [29]. The samples were shaken and brought back into darkness for 2 h. The absorbance readings were performed at 765 nm. Results were expressed as mg of gallic acid equivalent (mg GAE) g^−1^ of dried açaí.

Antioxidant evaluation of extracts was carried out through three antioxidant assays: Trolox Equivalent Antioxidant Capacity (TEAC), 2,2-diphenyl-1-picrylhydrazil (DPPH), and Ferric Reducing Antioxidant Power (FRAP). All antioxidant results were expressed as mg Trolox g^−1^ dried açaí. Both TEAC and DPPH methods measure the antioxidant power of extracts by the percentage inhibition of ABTS^+•^ and DPPH^•^ radicals, respectively, via both single-electron transference (SET) and hydrogen-atom transference (HAT) mechanisms [30]. The cationic radical ABTS^+•^ was generated from an oxidation reaction of the ABTS reagent with potassium persulfate incubated in the dark at room temperature for 16 h. ABTS^+•^ working solution was obtained by dilution of the concentrated solution until an absorbance value of 1 at 734 nm. 3 mL of working ABTS^+•^ radical solution was mixed with 300 μL of each extract and three controls were prepared with the addition of 300 μL of water. DPPH radical-scavenging activity of açaí extracts was evaluated according to the method described by Okada and Okada with some modifications [31,32]. 5 mL of extracts were incubated with 0.5 mL of 6.4 × 10^−4^ DPPH solution for 30 min in the dark at room temperature, and absorbance was determined at 517 nm. FRAP assay measures the antioxidant activity through reduction of ferric 2,4,6-tripyridyl-s-triazine (TPTZ) to a colored product via SET mechanism. FRAP reagent was prepared by mixing 25 mL of 0.3 M acetate buffer (pH 3.6) with 2.5 mL of 10 mM TPTZ (2,4,6-tripyridyl-s-triazine) and 2.5 mL of 20 mM FeCl_3_. 2850 μL of FRAP reagent was mixed with 150 μL of each extract and the absorbance was measured at 593 nm after 30 min of reaction at room temperature. The assays were performed in triplicate and results were expressed as mg Trolox/g fruit.

### 2.3. Encapsulation of Açaí Extract with Highest Phenolic Content

#### 2.3.1. Determination of Zein–Açaí Extract Solution Properties

The açaí extract to be encapsulated (Aç2) was selected according to the highest concentration of active compounds by means of highest polyphenolic content and antioxidant capacities. Aç2 was obtained following the same procedure as described in Section 2.2.1 and subsequently, this extract was concentrated to a final concentration of 0.4 g dried açaí mL^−1^ using a rotary evaporator. This concentrated extract was named AÇ_CC_. Electrospinning solutions were prepared with 2 mL of AÇ_CC_, 8 mL of ethanol and zein was added at different concentrations (16, 18 and 20% *w v*^−1^). The mixtures were gently stirred at room temperature for 1 h until homogeneous solutions were obtained (ZN16-AÇ_CC_, ZN18-AÇ_CC, and_ ZN20-AÇ_CC_, respectively). Additionally, three control solutions of zein using 80% ethanolic solution were prepared at the same concentrations without the extract to study the effect of the incorporation of açaí extract on the properties of the polymer solutions (ZN16, ZN18, and ZN20, respectively).

The zein–açaí extract and control zein solutions were characterized by determination of viscosity and conductivity. Viscosity was evaluated using the SC4-18 spindle at a deformation rate of 79.2 s^−1^. In addition, the conductivity was measured using a conductivity meter from 0.01 to 1000 mS cm^−1^. Both studies were performed in triplicate at room temperature.

#### 2.3.2. Electrospinning Process of the Zein–Açaí Extract Solutions

The encapsulation was carried out using electrospinning equipment (Spraybase^®^power SupplyUnit, Maynooth, Ireland) with a vertical standard configuration equipped with a capillary connected to a high-voltage source. The technique was carried out at room temperature and 40% relative humidity. Initially, the purpose was to reveal the optimal concentration of zein to obtain electrosprayed capsules. In this process, the capillary was located 10 cm from the collector plate using a voltage of 13 kV. Each zein–açaí extract solution was introduced in a 5 mL syringe, which was expelled by the capillary during the process with an injection flow of 0.15 mL h^−1^. The first samples were collected on a slide for easy observation by optical microscopy, and to obtain an initial and fast appreciation of the morphology of the electrospun structures. Once the zein concentration was fixed, the electrospinning parameters were studied to be able to fix the best conditions to obtain electrosprayed particles through a homogeneous and stable process. Flow rate and distance between capillary and collector were studied as follows: Samples S1, S2, and S3 with 10 cm distance and flow rates 0.3, 0.4, and 0.5 mL h^−1^, respectively; and samples S4, S5, and S6 with 12 cm as distance and 0.3, 0.4, and 0.5 mL h^−1^, respectively.

### 2.4. Characterization of Electrosprayed Açaí-Containing Capsules

#### 2.4.1. Morphological Analysis 

The morphologies and size distribution of electrosprayed zein ZN/AÇ_EXT_ capsules resulting from processing samples S1–S6 were observed. Electrosprayed structures were previously coated with gold–palladium and analyzed by scanning electron microscopy (ZeissEVO MA10SEM, Oberkochen, Germany) at 20 kV. Average particle diameters were analyzed with Image analyzer software (Image J v 1.37) (Bethesda, MD, USA).

#### 2.4.2. Structural Analysis 

Functional chemical group analysis of the samples was performed through spectrometer equipment Bruker Alpha (Ettlingen, Karlsruhe, Germany) with transmission spectra accessory mode. Concentrated açaí extract was previously lyophilized for further analysis (AÇ_EXT_). Electrosprayed zein particles without açaí (ZNe) were also analyzed to study possible chemical interactions between both components. Pellets with samples and potassium bromide were prepared by pressure and the spectra were obtained in a range from 4000 to 400 cm^−1^ with a resolution of 2 cm^−1^ and 64 scans.

#### 2.4.3. Phenolic Loading Capacity (LC) and Encapsulation Efficiency (EE)

Loading capacity (%LC) was measured as the mass ratio between the total polyphenols content (PT) determined in the ZN/AÇ_EXT_ capsules and the theoretical phenolic content incorporated during the preparation of the capsules. 0.015 g of ZN/AÇ_EXT_ was dissolved with 1.2 mL of 80% ethanol for PT measurement. Samples were filtered through a 0.22 μm filter and the supernatant was analyzed following the Folin–Ciocalteu method described previously. It is worth mentioning that zein was also analyzed and did not present phenolic content interference. The encapsulation efficiency (%EE) was determined following the procedure of Idham, Muhamad & Sarmidi (2012) with some modifications [33]. This test consisted of the determination of total polyphenols (PT) and surface polyphenols (PS) of the encapsulated extract. 0.015 g of ZN/AÇ_EXT_ was weighed and mixed with 1.2 mL of distilled water in an Eppendorf tube to measure PS. On the other hand, 0.015 g of ZN/AÇ_EXT_ was mixed with 1.2 mL of 80% ethanol for PT measurement. In both cases, the samples were vortexed for 30 s. Finally, each sample was filtered through a 0.22 μm filter and the supernatants were analyzed following the Folin–Ciocalteu method described previously. %EE was calculated following Equation (1):% EE = (PT − PS) · PT^−1^ · 100(1)

### 2.5. Thermal Properties of Zein–Açaí Capsules

#### 2.5.1. Thermal Stability Test

Thermogravimetric analyses (TGA) of açaí fruit (AÇ), lyophilized hydroalcoholic açaí extract (AÇ_EXT_), and encapsulated (ZN/AÇ_EXT_) were carried out using a Mettler Toledo Gas Controller GC20 Stare System (Schwerzenbach, Switzerland) TGA/DSC. Samples were heated from 30 to 600 °C at 10 °C/min under nitrogen atmosphere.

#### 2.5.2. Stability of Total Phenolic Content of AÇ by Encapsulation 

Phenolic content of AÇ, AÇ_EXT_ and ZN/AÇ_EXT_ samples were determined before (at room temperature) and subsequently exposed to high-temperature conditions. These parameters were selected in order to simulate common heat-treatment processes that these active capsules could suffer when being used as nutraceuticals incorporated into food: (A) sterilization process: autoclaving at 121 °C and 15 psi for 15 min; and (B) baked process: 180–185 °C for 25 min. 

### 2.6. In Vitro Digestion

Dispersions of açaí (AÇ), lyophilized hydroalcoholic açaí extract (AÇ_EXT_), and encapsulated (ZN/AÇ_EXT_) were subjected to in vitro digestibility assays to evaluate their bioaccesibilities through the analysis of phenolic content after a gastric and intestinal phase. In vitro gastric and intestinal phases were prepared based on the consensus of the protocol for simulating static digestion method described by the COST Action InfoGest [34]. Simulated gastric fluid (SGF) consisted of a stock gastric solution (6.9 mM KCl, 0.9 mM KH_2_PO_4_, 25 nm NaHCO_3_, 47.2 mM NaCl, 0.12 mM MgCl_2_*6H_2_O, 0.5 mM 2NH_4_CO_3_), water, 0.3 M CaCl_2_, 1.0 M HCl, lecithin, and pepsin. At this stage, the different dispersions were mixture with SGF in a 1:1 proportion and incubated at 37 °C for 90 min with continuous agitation at 200 rpm, where pH of the mixture was monitored and controlled to a value of 2 through the addition of 0.5 M HCl, and by using an automatic titrator (902 Titrando, Metrohm, USA). Simulated intestinal fluid (SIF) consisted of a stock gastric solution (6.8 mM KCl, 0.8 mM KH_2_PO_4_, 85 nM NaHCO_3_, 38.4 mM NaCl, 0.33 mM MgCl_2_.6H_2_O), water, 0.3 M CaCl_2_, 1.0 M NaOH, bile, pancreatin, and lipase. At this stage, the mixtures obtained from the gastric phase were mixed with SIF in a 1:1 proportion and incubated at 37 °C for 120 min with continuous agitation at 200 rpm. During this second stage, pH was monitored and maintained at pH 7 by adding 0.5 M NaOH. The resulting aliquots after each stage of in vitro digestion were collected and frozen at −18 °C, and phenolic content were evaluated following the Folin–Ciocalteu method. All experiments were carried out in duplicate.

### 2.7. Statistical Analysis

One-way analyses of variance were carried out. The software SPSS version 11.5 (SPSS Inc., Chicago, IL, USA) was used. Differences in pairs of mean values were evaluated by the Tukey b-test at a confidence interval of 95%. Data were represented as the average ± standard deviation. 

## 3. Results and Discussion

### 3.1. Evaluation of Polyphenolic Content and Antioxidant Capacity of Açaí Extracts

Total phenolic content (TPC) and antioxidant capacities of açaí fruit extracts measured through Folin–Ciocalteu, TEAC, DPPH, and FRAP methods are listed in Table 1. The extraction of natural active compounds was highly influenced by the solubility of these compounds in the extractive solvent. In this case, hydroalcoholic extraction achieved the greatest performance by extracting the highest amount of phenolic and antioxidant compounds with different polarities related to both ethanol and water [35]. A better solvation of the compounds, as a result of the hydrogen bond interactions between the polar sites of the antioxidant molecules and both solvents, in comparison to whether each solvent is used separately for extraction, has been demonstrated [36]

The Folin–Ciocalteu method is broadly used to measure the content of total phenolic compounds in plant products. It is based on the fact that the phenolic compounds react with the Folin–Ciocalteu reagent at basic pH, giving rise to a blue coloration that can be easily spectrophotometrically determined [37]. The resulting TPC values indicated that the type of solvent directly affected the total number polyphenols extracted. Other works have shown lower TPC values for açaí fruit pulp extracts, e.g., 31.7 ± 0.6 and 26.7 ± 0.5 mg GAE g^−1^ açaí when extractions were carried out by using 1% acetic acid aqueous solution and a solvent mixture acetone/water 70:30, respectively [36,38]. Açaí fruit has been revealed to be the fruit with the highest total polyphenolic content, followed by murtilla, calafate, and maqui, whose TPC values were 34.9, 33.9, and 31.2 mg GAE g^−1^, respectively, according to the antioxidant database directed by Speisky et al. [39]. Nevertheless, a reliable comparison of TPC values between studies is very difficult to achieve because the lack of standardization of this assay and the extraction conditions can imply several orders of significant difference in detected phenols [30].

The highest phenolic content and antioxidant capacity of açaí fruit pulp was obtained for the Aç2 extract, except for the TEAC assay, where the hydroalcoholic extract Aç2 and aqueous Aç3 values did not present significant differences. This fact was due to the scavenging capacity of the compounds that were extracted and their affinity with the radical. This assay demonstrated that radical ABTS^+•^ presented great affinity for hydrophilic systems [40], principally aqueous systems, and very poor affinity for ethanolic extraction. The antioxidant capacity of extracts is the expression of the different phenolic components, which behave through different mechanisms of interactions with oxidative species. Therefore, it is necessary to perform more than one antioxidant method to reflect both lipophilic and hydrophilic capacities. TEAC assay is based on the generation of the cationic radical ABTS^+•^ with blue–greenish coloration, which is applicable to both hydrophilic antioxidant systems as to the lipophilic ones, while the DPPH assay uses the radical DPPH^•^ dissolved in organic medium such as ethanol and, therefore, is more applicable to hydrophobic systems [41]. Thus, in agreement with Floegel et al. (2001), which compared both antioxidant methods by measuring antioxidant activities of different groups of fruit, vegetables, and beverages, TEAC results were higher than DPPH values [42]. On the other hand, FRAP assay revealed that Aç2 presented the highest antioxidant activity measured through SET. Schauss et al. (2006) also demonstrated that antioxidant activity of this fruit measured through ORAC method resulted in the highest reported scavenging activity values for a fruit or berry [43].

### 3.2. Characterization of the Zein Extract Solutions

Zein concentration of electrospinning solutions was one of the main parameters that determined the morphology of the fibers or capsules because they directly affected the viscosity and the conductivity values. Table 2 shows viscosity and conductivity results of zein solutions. As expected, viscosity significantly increased as the concentration of zein increased because this parameter is closely related to the polymeric chain entanglement and intercalation [44].

The incorporation of the açaí extract significantly increased the viscosity owing to the presence of more solutes in the solution, which could increase entanglements of zein protein molecules [45]. When the highest concentration of zein (20% *w*/*v*) was processed, the chains remained entangled enough to resist the electric charges that tend to break the jet during the electrospinning process, resulting in fiber formation. On the other hand, when using 16% (*w*/*v*) zein solution, the electric charge broke the chains, resulting in dispersed droplets whose evaporation originated spherical particles or capsules (see Appendix A), thereby being the zein concentration selected.

The conductivity values of the solutions slightly decreased with increasing zein concentration and significantly increased when the extract was added. Possibly, this fact could be related to the presence of polyphenols solved in the hydroalcoholic extract that enhanced the conductivity [46]. When conductivity increased, the difference in electric charges between the Taylor cone and the collector plate increased, which promoted capsule development during the formation of droplets [44].

### 3.3. Morphological Studies of Açaí-Containing Zein Capsules

Once the zein concentration was fixed, samples S1–S6 were processed through an electrospinning process to obtain particles with great homogeneity. The size and homogeneity are highly dependent on electrospinning process parameters. The particle size distribution plots and Scanning Electronic Microscopy (SEM) micrographs of the samples at different electrospinning encapsulation conditions are shown in Figure 1. SEM micrographs of the electrosprayed zein–açaí capsules demonstrated particles with cavities and similar shape to those found in previous works [47,48].

Samples S1, S2, S4, and S6 were found to have a smaller average particle size: 882, 924, 899, and 896 nm, respectively, without significant differences between them. According to Duque et al. (2013), when increasing the injection flow, there is less solvent evaporation time, so there can be agglomerations of droplets and an increase in the diameter of particles [44]. This fact occurred with samples S1 and S3, where, by increasing the flow from 0.3 to 0.5 mL h^−1^, the particle average diameter increased from 0.88 to 1.13 µm. On the other hand, the distance between the tip of the capillary and the collector plate also influenced on the homogeneity and morphology of the capsules. The sample with the highest homogeneous particle size, based on the smaller standard deviation, turned out to be sample S2. Hence, the parameters selected were 0.4 mL h^−1^ flow rate and 10 cm height. 

### 3.4. Structural Characterization 

The infrared spectra of electrosprayed zein particles (ZN_e_), freeze-dried açaí extract (AÇ_EXT_) and electrosprayed zein-containing açaí (ZN/AÇ_EXT_) are displayed in Figure 2. Zein particles obtained through the electrospinning process presented characteristic peaks of zein protein at 3309 cm^−1^ which represented the N–H stretching of amide A, and peaks at 2966 and 2872 cm^−1^ derived from the stretching of the C–H aliphatic groups. The peak at 1655 cm^−1^ represented the stretching vibration of C=O group from amino acids and the bands at 1543 and 1450 cm^−1^ illustrated the flexion vibration of N–H and vibrational stretching of the C–N peptide bonds, respectively [49,50].

Lyophilized açaí extract spectra exhibited a similar pattern to other berries, such as maqui and murta [51,52]. A broad sign with peak at 3383 cm^−1^ could represent some hydroxyl groups O–H and aliphatic C–H from the polyphenolic compounds. The peak centered at 1615 cm^−1^ was assigned to the C=C vibrations from aromatic systems. The region between 1500 and 1340 cm^−1^ (centered at 1413 cm^−1^) represented the deformation vibrations of phenolic O–H groups. A peak was distinguished in the region of 1150 and 1040 cm^−1^, which was attributed to C–O stretching vibrations [53]. Although ZN/AÇ_EXT_ presented similar bands to ZNe spectra, a certain displacement of these peaks was observed. This fact indicated the presence of a certain intermolecular interaction between zein protein and phenolics from açaí fruit extract. Zein protein, which contains mostly non-polar amino acids, favored chemical interactions with phenolic functional groups, increasing the protection of active compounds. Non-covalent hydrophobic interactions and hydrogen bonds were probably the main mechanisms of interaction between zein and polyphenols [54,55].

### 3.5. Loading Capacity (LC) and Encapsulation Efficiency (EE)

The encapsulation efficiency (EE) concept has given rise to different definitions in various works relating to compound encapsulation. In this study, EE was defined as the precise amount of active compounds that was actually protected in the capsule. On the other hand, the performance of the encapsulation process was distinguished as loading capacity (LC). Loading capacity (%LC) resulted in (98.6 ± 1.6)% value, since theoretical phenolic content of ZN/AÇ_EXT_ capsules was 2104 mg gallic acid g^−1^, and PT value after dissolution of ZN/AÇ_EXT_ was 2075 mg gallic acid g^−1^ ZN/AÇ_EXT_. This value justified the efficiency of this technology to encapsulate bioactive compounds without compromising its activity. Similar loading capacity values between 85% and 95% were observed to encapsulate other natural extracts by using the electrospinning technique, such as green tea extract in zein and carotenoids from tomato peel extract in gelatin [45,48]. 

Encapsulation efficiency value is essential to study the number of active compounds trapped in the capsule and the ability of the material to retain them [56]. EE value indicated (72.1 ± 1.7)% of phenolic content from açaí fruit extract was efficiently encapsulated. EE depends to a large extent on the affinity between the polymer matrix and active compounds. During the encapsulation process, the açaí fruit extract as core material was mixed with the substance of zein and the generated droplets were solidified by the evaporation of the ethanol and water [57]. Yao et al. (2016) also indicated that the EE would also be influenced by variations in the morphology that arise in the fibers or capsules due to the concentration of the solution and process conditions. In this study, the result indicated that more than 70% of the extract was efficiently trapped and distributed inside the capsule [58]. A similar value was found in the Flores et al. study of physical and storage properties of cranberry pulp encapsulated in whey protein by spray drying [59]. In addition, it is important to consider the methodology used to determine the efficiency, since EE results depend greatly on the methodology and, principally, the solvents used for the extraction of components.

### 3.6. Thermal Studies of Zein-Containing Açaí Extract Capsules

#### 3.6.1. Thermal Stability Test

Figure 3 shows the thermogravimetric curves (TGA) (Figure 3A) and their respective derivatives (DTGA) (Figure 3B) of the samples. An initial stage of weight loss between 30 and 100 °C was observed for all the compounds, which indicated a loss of water and some volatile compounds. The thermogram of the electrosprayed zein particles presented a second degradation process between 270 and 450 °C, with a peak of maximum degradation at 332.8 °C. This second stage is attributed to the main degradation of the protein, causing changes in the structure due to the breakdown of low-energy intermolecular bonds that maintain their conformation [26,27,60]. Lyophilized açaí fruit extract presented an early degradation that started at approximately 100 °C, showing a maximum degradation at 162.5 °C. This extract was mainly composed of anthocyanins, flavonoids highly sensitive to temperature [16]. The anthocyanins of the extract were totally unprotected and exposed to degradation due to the increase in temperature. When the extract was encapsulated, the thermogram presented a similar degradation profile to zein. The incorporation of açaí fruit extract did not affect the protein stability. The degradation of açaí fruit was not exhibited because zein effectively protected the extract, delaying its degradation.

The main degradation of dehydrated fruit AÇ occurred at higher temperatures, approximately at 180–190 °C, and displayed two peaks of maximum degradation at 320.5 and 406 °C. This fact can be possibly explained because the dehydrated fruit contained a food matrix based on husk and pulp that could exert some protection to the active compounds, while the extract is a concentrated sample of antioxidants totally exposed to heating [12,61].

#### 3.6.2. Thermal Protection of Açaí Phenolic Compounds Encapsulated in Zein

The thermal stability of the encapsulated extract was also evaluated by determining the loss of polyphenols when exposed to two thermal treatments of high-temperature processing: sterilization and baking. In addition, the stability of the phenolic content of dehydrated açaí fruit (AÇ) and lyophilized açaí extract (AÇ_EXT_) was also analyzed. Figure 4 shows the loss of the phenolic content after each heat treatment.

In the case of the commercial dehydrated açaí fruit, a phenolic content reduction greater than 40% was displayed after both thermal processes without significant differences between both treatments. In the case of the encapsulated extract, phenolic content loss values were 5% and 20% approx. after sterilization and baking, respectively. Encapsulated açaí phenolic compounds presented a greater stability against both treatments compared to two other samples of açaí fruit, principally during a sterilization simulation process. Otherwise, during the baking process, the amount of phenolic loss correlated with the amount of phenols on the surface (shown in Section 3.5) which were the phenolic content available to degradation. This fact confirmed the protective effect of the encapsulation in zein by the electrospinning technique, which was clearly demonstrated when comparing with AÇ_EXT_ phenolic loss. The sterilization process (121 °C) was a lesser influence than the baked (180 °C) over lyophilized açaí extract, exhibiting phenolic content reductions of 10 and 55%, respectively. Although sterilization did not cause a large degradation rate, in the case of baking, it turned out to be the sample that suffered the highest phenolic decrease, possibly due to higher temperature and longer exposure time. This fact is in agreement with AÇ_EXT_ TGA thermogram (Figure 3B) that indicates AÇ_EXT_ present an early degradation that starts at 100 °C and the maximum degradation temperature occurs at approximately 160 °C. Thus, the sterilization process clearly affects the phenolic content to a lesser extent than baking (180 °C), which occurs at a higher temperature than the maximum degradation. Other works have already shown that temperature is a determining factor in the degradation of polyphenols. Pacheco et al. studied the phytochemical composition and thermal stability of two commercial açaí species and concluded that the changes in antioxidant capacity during warming were highly related to the loss of anthocyanins because their polyphenols, such as phenolic acids and flavone glycosides, were not significantly altered [62]. The thermal degradation of anthocyanins can lead to a variety of species depending on the severity and nature of the heating. High temperature causes losses in the glycosidant sugar of the molecules and the opening of the ring, producing the so-called “colorless chalcones”.

### 3.7. In Vitro Bioaccesibility Study

Phenolic content (PC) of AÇ, AÇ_EXT_, and ZN/AÇ_EXT_ were determined after each stage of in vitro digestion (gastric and intestinal stages) to evaluate phenolic content release within gastrointestinal tract (GI) and to assess bioaccessibility at the end of digestion process. Bioaccessibility can be defined as the fraction of a compound that is soluble in the gastrointestinal (GI) tract that is available for absorption, or the fraction of a compound released from its matrix in the GI tract [63]. Therefore, in this study, bioaccessibility was defined as phenolic content recovered from the intestinal phase after in vitro digestion. Table 3 shows phenolic content of gastric and intestinal stages per gram of each sample. Thus, significant differences to these values were because of their intrinsic phenolic content between samples. In general, polyphenols from açaí samples without encapsulation showed a moderate stability under gastric conditions since the phenolic content decreased between 20% and 35% with respect to intrinsic content (Table 3).

As with during thermal analyses (Section 3.6.1), pulp matrix from açaí fruit generated some protection to açaí polyphenols during the gastric digestion phase. Other works have also shown flavonoid oligomers were degraded to smaller units at low pH values [64]. These results agreed with Gullón et al. (2015) who observed that the total phenolic recovery from pomegranate peel flour (35.8%) decreased after gastric digestion. Results after intestinal stage demonstrated phenolic content depended on açaí sample. Although AÇ polyphenols degraded to a lesser extent during the gastric phase, the dried fruit pulp showed the greatest PC decrease after intestinal digestion phase, remaining approx. 20% from initial phenolic content. On the other hand, although AÇ_EXT_ suffered a higher reduction after gastric phase, the total PC loss was shorter than AÇ, with a PC reduction close to 60% with respect to non-digested samples. In general, these results suggested that several changes in phenolic compounds as a chemical structure modification, reduction of their solubility due to pH, and/or interaction with other compounds might have occurred during the duodenal stage [65,66]. On the other hand, encapsulated zein-containing açaí presented an interesting behavior because phenolic content values increased after both in vitro digestion processes. During the gastric phase, the phenolic components released from the zein capsules was approx. 60% with respect their intrinsic content, and this value increased after the intestinal stage. This fact is probably due to the breakage of the zein structures, thanks to the digestion of protein matrix, allowing the release of polyphenols. Gómez-Mascaraque et al. (2019) also revealed an increase to antioxidant capacity derived from catechin from zein and gelatin electrosprayed systems. 

## 4. Conclusions

Polyphenolic content and antioxidant activity studies have demonstrated açaí to be the fruit with the highest content of active compounds. Studies of electrospinning encapsulation for the development of zein capsules containing hydroalcoholic açaí extract have also indicated the protective effect of these protein structures. This fact was certainly due to the positive chemical interactions observed through infrared spectroscopy between protein and active compounds from açaí extract. Zein was confirmed to be an adequate protein for the encapsulation of thermal sensitive active compounds by improving the thermal stability of polyphenols from açaí fruit when exposed to high-temperature treatments related to processed foods. After in vitro digestion processes, açaí polyphenols were also protected and diffused thanks to the breakage of zein protein electrosprayed capsules.

## Figures and Tables

**Figure 1 antioxidants-08-00464-f001:**
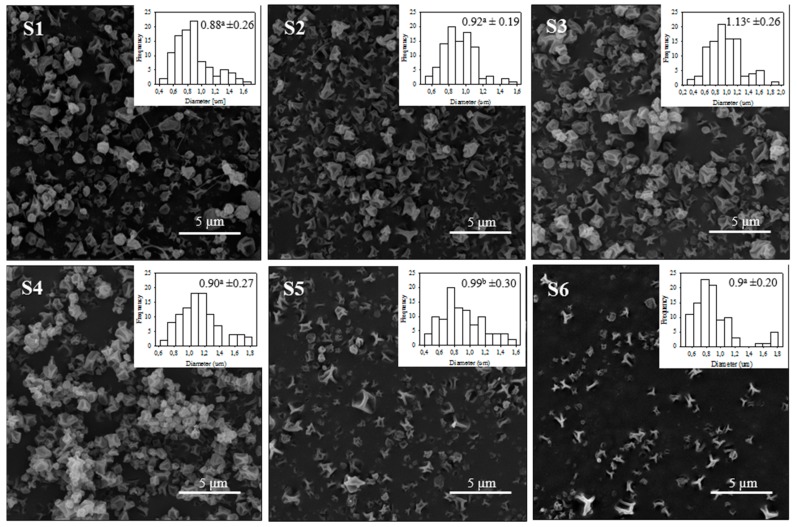
Histograms of electrosprayed zein–açaí structures with size distribution information as average and deviation (μm) and SEM micrographs of samples: S1, S2 and S3) 10 cm height and flow rates 0.3, 0.4, and 0.5 mL h^−1^, respectively; and S4, S5, and S6) 12 cm height and flow rate 0.3, 0.4, and 0.5 mL h^−1^, respectively. Values a, b, and c indicate significant differences between the samples, determined through a one-way analysis of variance (ANOVA) (*p* < 0.05). SD corresponds to the standard deviation.

**Figure 2 antioxidants-08-00464-f002:**
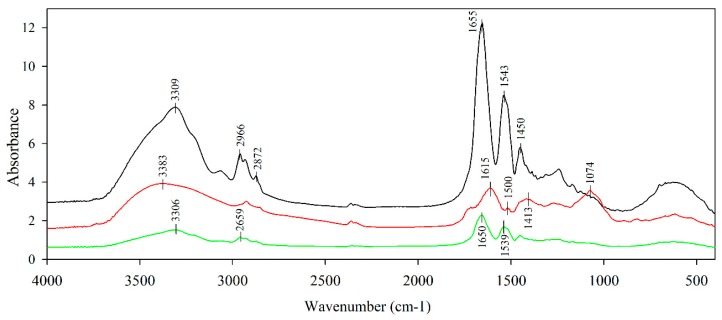
FTIR spectra of: electrosprayed zein particles, ZNe (black line); açaí extract, AÇ_EXT_ (red line); and electrosprayed zein particles containing açaí, ZN/AÇ_EXT_ (green line).

**Figure 3 antioxidants-08-00464-f003:**
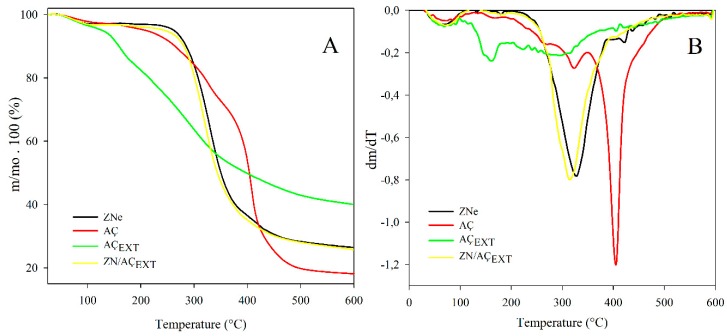
(**A**) Weight loss; and (**B**) derivative of the weight loss of electrosprayed zein (Zne), dried açaí fruit (AÇ), lyophilized açaí fruit extract (AÇ_EXT_), electrosprayed zein capsules containing açaí (ZN/AÇ_EXT_).

**Figure 4 antioxidants-08-00464-f004:**
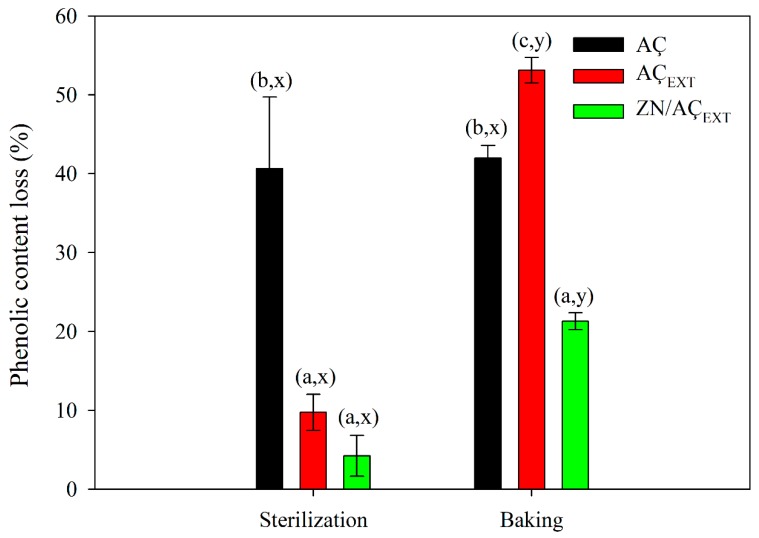
Loss of polyphenolic content (%) of the samples when subjected to thermal treatments. Values a, b, and c indicate significant differences between samples for the same thermal treatment. On the other hand, x and y indicate significant differences between thermal treatments for the same sample, determined through a one-way analysis of variance (ANOVA) (*p* < 0.05).

**Table 1 antioxidants-08-00464-t001:** Polyphenolic content and antioxidant capacities results of açaí fruit extracts.

Extract	TPC	TEAC	DPPH	FRAP
	(mg GAE/g)	(mg Trolox/g)	(mg Trolox/g)	(mg Trolox/g)
Aç1	23.8 ^a^ ± 0.2	26.1 ^a^ ± 0.9	14.1 ^a^ ± 0.1	32.8 ± 0.7
Aç2	43.4 ^c^ ± 0.2	130.1 ^b^ ± 0.9	62.7 ^c^ ± 0.5	68.4 ^c^ ± 0.3
Aç3	35.4 ^b^ ± 0.4	122.4 ^b^ ± 2.0	51.2 ^b^ ± 1.2	57.2 ^b^ ± 0.6

Letters a–c indicate significant differences among the extracts of the same method.

**Table 2 antioxidants-08-00464-t002:** Viscosity and conductivity of polymer solutions with and without açaí extract used during electrospinning process.

Zein concentration	Viscosity (cP)		Conductivity (mS cm^−1^)	
(%, *w*/*v*)	ZN	ZN-AÇ_CC_	ZN	ZN-AÇ_CC_
16	18.2 ^a,x^ ± 0.3	20.6 ^a,y^ ± 0.2	694 ^b,x^ ± 4	750 ^b,y^ ± 6
18	21.9 ^b,x^ ± 0.2	26.7 ^b,y^ ± 0.6	685 ^a,x^ ± 1	717 ^a,y^ ± 1
20	27.2 ^c,x^ ± 0.9	30.0 ^c,y^ ± 0.3	655 ^a,x^ ± 3	716 ^a,y^ ± 7

Letters a–c indicate significant differences among the different zein concentration samples. Letters x, y indicate significant differences between sample with and without açaí extract at the same zein concentration.

**Table 3 antioxidants-08-00464-t003:** Phenolic content as (mg gallic acid per gram) of açaí samples after in vitro digestive processes.

Sample	Gastric	Intestinal
AÇ	3044 ^b,y^ ± 32	988 ^a,x^ ± 108
AÇ_EXT_	12,981 ^b,z^ ± 461	6462 ^a,z^ ± 402
ZN/AÇ_EXT_	1486 ^a,x^ ± 148	2963 ^b,y^ ± 58

Letters a, b indicate significant differences among the different digestive stages of a sample. Letters x, y indicate significant differences between samples during the same digestive process.

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
