# Peer review of "Enhancing Thermal Stability and Bioaccesibility of Açaí Fruit Polyphenols through Electrohydrodynamic Encapsulation into Zein Electrosprayed Particles"

_antioxidants, 2019, doi:10.3390/antiox8100464_

Round 1
Reviewer 1 Report
The work "Enhancing thermal stability and......" present good results and has a certain originality. In my opinion the authors should review the English language, moreover the authors should explain to me what " 600 microL of anydrous sodium carbonate solution.....was added" line 99-100, means and why a solution, after which the work can be considered for the publication in Antioxidants.
Author Response
Regarding to this evaluation, authors have revised methodology and some improvements were carried out. Specifications about some methods were included and the writing of the methodology was revised.
Comments and Suggestions for Authors
The work "Enhancing thermal stability and......" present good results and has a certain originality. In my opinion the authors should review the English language, moreover the authors should explain to me what " 600 microL of anydrous sodium carbonate solution.....was added" line 99-100, means and why a solution, after which the work can be considered for the publication in Antioxidants.
Authors thank the good comments.
English was carefully revised.
The Folin-Ciocalteu assay is a colorimetric method based on the reduction of Folin-Ciocalteu reagent (a mixture of sodium molybdate, sodium tungstate, and other reagents) through single electron transfer from phenol to complexed Mo (VI) (Mo(VI) + e à Mo(V)). In alkaline medium (pH˜10 adjusted by a sodium carbonate solution), the dissociation of a phenolic proton leads to a phenolate anion that is able to reduce the F-C reagent. The result of the reaction between phenolic compounds and the F-C reagent is the development of blue complex, which presents wavelength of maximum absorption close to 760 nm (Lamuela-Raventós, R., 2017).
Sodium carbonate is added to create basic pH because the fast oxidation reaction of phenols occurs by the usage of alkali. And the most commonly used alkali is sodium carbonate.Lamuela-Raventós, R. M. (2017). Folin-Ciocalteu method for the measurement of total phenolic content and antioxidant capacity. Measurement of Antioxidant Activity & Capacity, 107–115.
This reference was included in case readers want to deep on this method (see Ref. 29).

Reviewer 2 Report
Line 22. “and” should be “with”
Line 23. Suggest “Fourier transform infrared spectroscopy”
Line 25. Suggest dropping “was obtained.” It is inferred as results were obtained (70%).
Line 32-34. The authors have the opportunity in the introduction to cite a paper that not only demonstrated experimentally that freeze-dried acai pulp significantly reduced the risk for atherosclerosis but also described its mechanisms of action. See: Xie C, Kang J, Burris R, Ferguson ME, et al. Açaí juice attenuates atherosclerosis in ApoE deficient mice through antioxidant and anti-inflammatory activities. Atherosclerosis, 2011; 216: 327-333.
Further, the authors could then mention that the same collaborative group determined that through characterization and further elucidation of the pulp, they isolated a class of flavones that contained potent anti-inflammatory compounds. See: Xie C, Kang J, Li Z, et al. The açai flavonoid velutin is a potent anti-inflammatory agent: Blockade of LPS-mediated TNF-α and IL-6 production through inhibiting NF-κB activation and MAPK pathway.Journal of Nutritional Biochemistry, 2012; 23(9): 1184-1191.
Among the flavones were some particularly potent compounds with pronounced anti-inflammatory bioactivity such as the flavone, velutin. See: Xie C, Kang J, Zhimin L, Nagarajan S, et al. Velutin reduces lipopolysaccharide-induced proinflammatory cytokine TNFα and IL-6 production by inhibiting NF-κB activation. FASEB Journal, 2011, 25: 772.13.
By demonstrating the authors familiarity with the literature and the foundational studies that support their assertion that foods such as Acai may have beneficial health supporting properties worthy of interest, the reader is presented with compelling reasons to continue to read the findings of the authors. In terms of protecting human endothelial cells, the recent paper should be considered by Soares E, et al. Up-regulation of Nrf2-antioxidant signaling by Açaí (Euterpe oleracea Mart.) extract prevents oxidative stress in human endothelial cells. J Functional Foods, 37, 107-115, 2017.
Along the same lines, the authors mention that nutraceuticals may have therapeutic effects on health disorders related to neurodegeneration. Here again, the authors have an opportunity to demonstrate their familiarity with the literature by citing studies that have demonstrated thepotential health benefits related to neurodegenerative diseases. Some more recent studies related to Acai and its potential benefit to the brain and neurological functioning, especially related to aging, and given that Acai is the subject of the author’s manuscript, the authors should consider the papers by: Poulose SM, Fisher DR, Bielinski DF, Gomes SM, et al. Restoration of stressor-induced calcium dysregulation and autophagy inhibition by polyphenolics-rich acai (Euterpe spp.) fruit pulp extracts in rodent brain cells in vitro. Nutrition, 2014; 30(7): 853-862; Poulose SM, Fisher DR, Larson J, Bielinski DF, et al. Attenuation of inflammatory stress signaling by açai fruit pulp (Euterpe oleracea Mart.) extracts in BV-2 mouse microglial cells. Journal of Agricultural and Food Chemistry, 2012; 60: 1084-1093; Carey AN, Miller MG, Fisher DR, Bielinski DF, et al. Dietary supplementation with the polyphenol-rich acai pulps improves cognition in aged rats and attenuates inflammatory signaling in BV-2 microglial cells. Nutritional Neuroscience, 20(4), 238-245, 2017; The effect of acai (Euterpe spp.) fruit pulp on brain health and performance. In: Bioactive Nutraceuticals and Dietary Supplements in Neurological and Brain Disease: Prevention and Therapy. Watson RR, Preedy VR [Eds.]. Academic Press: Oxford, 2014, pp. 179-186.
Lines 34-37. Would suggest removing this sentence. Unless the authors relate this sentence specifically to Acai, it seems to come across as puffery, without specificity.
Lines 40-43. Acai is NOT a berry. It is a fruit. It may be similar in size to that of high-bush blueberries or raspberries, but its taxonomy makes the Acai palm fruit a fruit since it has a single seed. To date the literature has not reported a higher ROS scavenging activity in vitro for any fruit or berry than that of Acai.
Line 41. Replace “berries” with “Among fruits, it continues to show the most pronounced antioxidant bioactivity based on in vitro assays, or any fruit.”
Lines 41-43. Suggest the authors encourage readers to learn more about the range of bioactivities and properties shown for the fruit by reading the review chapter: Odendaal AY, et al. Potent antioxidant and anti-inflammatory flavonoids in the nutrient-rich Amazonian palm fruit, açaí (Euterpespp.). In: Polyphenols in Human Health and Disease.Oxidation and Antioxidant Activity of Polyphenols. Watson RR, Reedy VR, Zibadi S [eds.] Academic Press: San Diego, 2014, pp. 219-239.
Line 44. Replace “berry” with “fruit”
Lines 45-46. The attribution supporting this sentence should include the first authors to report its chemistry based on validated and replicated analytical outcomes: Journal of Agricultural and Food Chemistry, 2006; 54(22); 8598–8603; Journal of Agricultural and Food Chemistry, 2006; 54(22); 8604–8610. These two papers determined that much of the earlier literature on the pulp’s chemistry was not reproducible and or employed archaic methods and instrumentation.
Line 46. Besides being added to food, there is extensive availability of the pulp in the form of spray-dried and lypholized powders due to their extended shelf-life, as found in dietary supplements or beverages labeled as a dietary supplement. See: Jensen GS, Patterson KM, Barnes J, Carter SG, et al. In vitroand in vivo antioxidant and anti–inflammatory capacity of an antioxidant–rich fruit and berry juice blend. Results of a pilot and randomized, double–blind, placebo–controlled, crossover study. Journal of Agricultural and Food Chemistry, 2008; 56(18): 8326–8333.
Line 46-49. Need “do” as in “do not include …” The authors need to define what they mean by “high thermal processes.” Would this include flash pasteurization? Would that constitute “high”? Readers in food science would argue it is not. Blanching or cooking of the fruit certainly would be. The authors should show data demonstrating the degree stability lost by whatever “high” methods they contend degrades such polyphenolics as the anthocyanins, and which ones. A food scientist would not dispute hydrolyzing molecules at high temperatures would lead to “degradation” but the authors would need to cite papers that show this leads to their loss of functional properties, particularly in reference to fruits, or Acai specifically, if such studies exist.
Line 51. Would suggest the authors not use the word “suffering” to describe the digestive process. It serves a life-saving property in killing pathogens and along with breaking down foods into component parts that can be exposed to various enzymes essential to benefiting from their ingestion.
Line 51-53. Suggest avoiding the perception that the authors believe foods encapsulated are superior to un-encapsulated food stuffs. Humans have managed to get their nutrition from foods without encapsulation for hundreds of thousands of years. What the authors are offering the reader are alternative delivery systems that may or may not be advantageous over traditional means of securing the nutritional components found in foods. Suggest the paper do less to “sell” the reader on encapsulation, instead just show the data and let the food scientist and others determine if there is sufficient preferential evidence to suggest that encapsulation may warrant consideration given its added processing and cost. Note in line 52 the authors use the word, “essential”, to make the grand leap that encapsulation is superior to simply ingesting or masticating a food and deriving its nutritional and chemical composition.
Lines 57-71. All of what the authors state is supported by the literature. However, what is neglected is the evidence that utilization of electospinning and/or electrospraying foods results in the same experimentally demonstrated health benefits compared to more conventional methods of processing fruits, such as convection drying, lypholization, spray-drying, microwave-vacuum drying, etc.
Line 71. Reminder, it is a fruit, not berry.
Line 103-105. Curious why authors excluded using the ORAC assay? That large database developed by USDA from 1995-2010 allows for comparison of scavenging capacities. Total ORAC allows for evaluation of key radicals associated with disease progression, such as superoxide anion which produces the most prolific free-radical in the body, the peroxyl radical. What can TEAC, DPPH, and FRAP teach us about such radical chain reactions? ORAC represents a hydrogen atom transfer (HAT) reaction mechanism that is most relevant to human biology. See: Prior RL, Wu X, Schaich K. Standardized methods for the determination of antioxidant capacity and phenolics in foods and dietary supplements. J Agric Food Chem. 53, 4290-4302, 2005. Also, particularly important to read is the paper by Prior found at: https://brunswicklabs.com/wp-content/uploads/2017/02/a_response_to_the_usda_orac_statement.pdf
Line 114. What modifications were made? Important to know as any modification of the method if carried out correctly could affect results.
Line 122. What does “richest” mean? Suggest using a scientific term, as this sounds like puffery.
Line 217. Which version?
Line 236/ Throughout the manuscript, the authors should replace “acai-berry extract” with “acai-fruit pulp extract”, as that is the source material for the author’s study. See line 445, where “berries” is used; wherever reference to acai as a berry is found in the submission, replace it with the word “fruit.”
Line 238-240. Neither calafate or maqui come close to the published ROS scavenging activities reported for acai in the literature. The authors may be relying on the paper by Speisky H, et al, for their opinion. From a commercialization standpoint it is understandable why this misinformation wouldt be perpetuated by promoters of these two food products, given how consumers are unfamiliar with who to interpret ORAC values. However, a careful review of the literature will reveal that lyophilized-acai-pulp has greater ROS scavenging activity. The ORAC for acai is reported; see: Antioxidant capacity and other bioactivities of the freeze–dried Amazonian palm berry, Euterpe oleraceaeMart. (Açai). Journal of Agricultural and Food Chemistry, 2006; 54(22); 8604–8610. The authors are likely relying on the paper by Speisky H, et al, for their opinion. As the authors will see on page 8,607 of the paper in J Agric Food Chem, the combined value for hydrophilic and lipophilic scavenging activity for lypohilized-acai is 1,027 umol of Trolox per gram. More significantly, the SOD scavenging is reported at 1,614 - the highest reported value for a fruit or berry.
Ultimately, what is important is evidence of demonstrated experimental evidence of a food’s beneficial health effects. A current review of the literature on Acai shows significantly more in vivo evidence of its benefits than for maqui, and certainly for calafate, whose production is limited and yet to become a significant exported food for Chile or Argentina.
Line 415-416. Suggest the authors include information on the short-comings (limitations) of relying on this method to evaluate bioaccessibility and explain how the term bioaccessibility is different from that of bioavailability.
Lines 445-454. After reading the submission and re-reading it a second time, one continues to search for what the compelling evidence is of the superiority for encapsulation especially given the lack of comparative data to other methods of preservation.
Authors should carefully review each reference. For example, on line 483, the word “comercial” is misspelled. Also, use of lower and upper case for titles of articles should be consistent and according to author guidelines.
Author Response
Regarding to this evaluation, English was revised and authors have carried out modifications in the introduction, methodology, results and discussion to improve the manuscript.
Comments and Suggestions for Authors
Line 22. “and” should be “with”. Corrected
Line 23. Suggest “Fourier transform infrared spectroscopy”. Changed.
Line 25. Suggest dropping “was obtained.” It is inferred as results were obtained (70%). Sentence was modified.
Line 32-34. The authors have the opportunity in the introduction to cite a paper that not only demonstrated experimentally that freeze-dried acai pulp significantly reduced the risk for atherosclerosis but also described its mechanisms of action. See: Xie C, Kang J, Burris R, Ferguson ME, et al. Açaí juice attenuates atherosclerosis in ApoE deficient mice through antioxidant and anti-inflammatory activities. Atherosclerosis, 2011; 216: 327-333.
Further, the authors could then mention that the same collaborative group determined that through characterization and further elucidation of the pulp, they isolated a class of flavones that contained potent anti-inflammatory compounds. See: Xie C, Kang J, Li Z, et al. The açai flavonoid velutin is a potent anti-inflammatory agent: Blockade of LPS-mediated TNF-α and IL-6 production through inhibiting NF-κB activation and MAPK pathway. Journal of Nutritional Biochemistry, 2012; 23(9): 1184-1191.
Authors agree with reviewer regarding to enrich discussion about acai properties to enhance significance of this fruit. Both studies were cited and references included (see Introduction and new References 10,12).
Among the flavones were some particularly potent compounds with pronounced anti-inflammatory bioactivity such as the flavone, velutin. See: Xie C, Kang J, Zhimin L, Nagarajan S, et al. Velutin reduces lipopolysaccharide-induced proinflammatory cytokine TNFα and IL-6 production by inhibiting NF-κB activation. FASEB Journal, 2011, 25: 772.13.
This reference was not included due to its high similatiry with Xie et al., 2012 that was included thanks to reviewer´s recommendation.
By demonstrating the authors familiarity with the literature and the foundational studies that support their assertion that foods such as Acai may have beneficial health supporting properties worthy of interest, the reader is presented with compelling reasons to continue to read the findings of the authors. In terms of protecting human endothelial cells, the recent paper should be considered by Soares E, et al. Up-regulation of Nrf2-antioxidant signaling by Açaí (Euterpe oleracea Mart.) extract prevents oxidative stress in human endothelial cells. J Functional Foods, 37, 107-115, 2017.
The authors appreciate the recommendations of studies related to the bioactivities of this fruit. Soares et al. (2017) was mentioned and included as Reference 6.
Along the same lines, the authors mention that nutraceuticals may have therapeutic effects on health disorders related to neurodegeneration. Here again, the authors have an opportunity to demonstrate their familiarity with the literature by citing studies that have demonstrated the potential health benefits related to neurodegenerative diseases. Some more recent studies related to Acai and its potential benefit to the brain and neurological functioning, especially related to aging, and given that Acai is the subject of the author’s manuscript, the authors should consider the papers by: Poulose SM, Fisher DR, Bielinski DF, Gomes SM, et al. Restoration of stressor-induced calcium dysregulation and autophagy inhibition by polyphenolics-rich acai (Euterpe spp.) fruit pulp extracts in rodent brain cells in vitro. Nutrition, 2014; 30(7): 853-862; Poulose SM, Fisher DR, Larson J, Bielinski DF, et al. Attenuation of inflammatory stress signaling by açai fruit pulp (Euterpe oleracea Mart.) extracts in BV-2 mouse microglial cells. Journal of Agricultural and Food Chemistry, 2012; 60: 1084-1093; Carey AN, Miller MG, Fisher DR, Bielinski DF, et al. Dietary supplementation with the polyphenol-rich acai pulps improves cognition in aged rats and attenuates inflammatory signaling in BV-2 microglial cells. Nutritional Neuroscience, 20(4), 238-245, 2017; The effect of acai (Euterpe spp.) fruit pulp on brain health and performance. In: Bioactive Nutraceuticals and Dietary Supplements in Neurological and Brain Disease: Prevention and Therapy. Watson RR, Preedy VR [Eds.]. Academic Press: Oxford, 2014, pp. 179-186.
The authors highly appreciate the recommendations of studies related to the bioactivities of this fruit. Most of the references were included and introduction was modified in order to enhance the importance and increase the interest by this fruit to the readers (See References 6-12).
Lines 34-37. Would suggest removing this sentence. Unless the authors relate this sentence specifically to Acai, it seems to come across as puffery, without specificity.
Authors have accepted the reviewer´s comment. Since the introduction has turned specifically to acai fruit, this sentence was no longer necessary.
Lines 40-43. Acai is NOT a berry. It is a fruit. It may be similar in size to that of high-bush blueberries or raspberries, but its taxonomy makes the Acai palm fruit a fruit since it has a single seed. To date the literature has not reported a higher ROS scavenging activity in vitro for any fruit or berry than that of Acai.
Authors thank reviewer´s correction, and several changes were made over the whole manuscript to correct them.
Line 41. Replace “berries” with “Among fruits, it continues to show the most pronounced antioxidant bioactivity based on in vitro assays, or any fruit.” Corrected
Lines 41-43. Suggest the authors encourage readers to learn more about the range of bioactivities and properties shown for the fruit by reading the review chapter: Odendaal AY, et al. Potent antioxidant and anti-inflammatory flavonoids in the nutrient-rich Amazonian palm fruit, açaí (Euterpespp.). In: Polyphenols in Human Health and Disease.Oxidation and Antioxidant Activity of Polyphenols. Watson RR, Reedy VR, Zibadi S [eds.] Academic Press: San Diego, 2014, pp. 219-239.
Reference included (Ref. 11).
Line 44. Replace “berry” with “fruit”. Replaced
Lines 45-46. The attribution supporting this sentence should include the first authors to report its chemistry based on validated and replicated analytical outcomes: Journal of Agricultural and Food Chemistry, 2006; 54(22); 8598–8603; Journal of Agricultural and Food Chemistry, 2006; 54(22); 8604–8610. These two papers determined that much of the earlier literature on the pulp’s chemistry was not reproducible and or employed archaic methods and instrumentation.
The first reference was included because the chemical composition of acai was extensively studied.
Line 46. Besides being added to food, there is extensive availability of the pulp in the form of spray-dried and lypholized powders due to their extended shelf-life, as found in dietary supplements or beverages labeled as a dietary supplement. See: Jensen GS, Patterson KM, Barnes J, Carter SG, et al. In vitroand in vivo antioxidant and anti–inflammatory capacity of an antioxidant–rich fruit and berry juice blend. Results of a pilot and randomized, double–blind, placebo–controlled, crossover study. Journal of Agricultural and Food Chemistry, 2008; 56(18): 8326–8333.
Sentence was corrected and extended.
Line 46-49. Need “do” as in “do not include …” Corrected
The authors need to define what they mean by “high thermal processes.” Would this include flash pasteurization? Would that constitute “high”? Readers in food science would argue it is not. Blanching or cooking of the fruit certainly would be. The authors should show data demonstrating the degree stability lost by whatever “high” methods they contend degrades such polyphenolics as the anthocyanins, and which ones. A food scientist would not dispute hydrolyzing molecules at high temperatures would lead to “degradation” but the authors would need to cite papers that show this leads to their loss of functional properties, particularly in reference to fruits, or Acai specifically, if such studies exist.
This part of Introduction was improved. A specific study of the exposure of acai berry at temperatures between 40 and 80 °C was mentioned (Ref. 17).
Line 51. Would suggest the authors not use the word “suffering” to describe the digestive process. It serves a life-saving property in killing pathogens and along with breaking down foods into component parts that can be exposed to various enzymes essential to benefiting from their ingestion.
Authors totally agree with reviewer. “Suffering” was deleted.
Line 51-53. Suggest avoiding the perception that the authors believe foods encapsulated are superior to un-encapsulated food stuffs. Humans have managed to get their nutrition from foods without encapsulation for hundreds of thousands of years. What the authors are offering the reader are alternative delivery systems that may or may not be advantageous over traditional means of securing the nutritional components found in foods. Suggest the paper do less to “sell” the reader on encapsulation, instead just show the data and let the food scientist and others determine if there is sufficient preferential evidence to suggest that encapsulation may warrant consideration given its added processing and cost. Note in line 52 the authors use the word, “essential”, to make the grand leap that encapsulation is superior to simply ingesting or masticating a food and deriving its nutritional and chemical composition.
The word “essential” was modified and sentence gained a different meaning regarding to the importance of encapsulation.
Lines 57-71. All of what the authors state is supported by the literature. However, what is neglected is the evidence that utilization of electospinning and/or electrospraying foods results in the same experimentally demonstrated health benefits compared to more conventional methods of processing fruits, such as convection drying, lypholization, spray-drying, microwave-vacuum drying, etc.
Authors understands the reviewer´s concern since the use of electrospraying is the important characteristic of this manuscript. The encapsulation of bioactive components through eletrospinning and electrospraying is a recent topic of research and the health benefits of products resulted from these techniques cannot be found yet. Nevertheless, research studies have clearly show electrospraying and electrospnning are techniques with functional advantages as sustained release property, high encapsulation efficiency and enhanced stability of encapsulated food bioactive compounds. A reference was included.
Line 71. Reminder, it is a fruit, not berry. Corrected
Line 103-105. Curious why authors excluded using the ORAC assay? That large database developed by USDA from 1995-2010 allows for comparison of scavenging capacities. Total ORAC allows for evaluation of key radicals associated with disease progression, such as superoxide anion which produces the most prolific free-radical in the body, the peroxyl radical. What can TEAC, DPPH, and FRAP teach us about such radical chain reactions? ORAC represents a hydrogen atom transfer (HAT) reaction mechanism that is most relevant to human biology. See: Prior RL, Wu X, Schaich K. Standardized methods for the determination of antioxidant capacity and phenolics in foods and dietary supplements. J Agric Food Chem. 53, 4290-4302, 2005. Also, particularly important to read is the paper by Prior found at: https://brunswicklabs.com/wp-content/uploads/2017/02/a_response_to_the_usda_orac_statement.pdf
Corresponding author knows very well Dr. Schaich´s work since she has worked with this amazing scientist at Rutgers University for six months. Her reference was already included in the manuscript and it is clearly evidence its importance. Dr. Schaich confirms a single antioxidant method underestimates and lacks of understanding the antioxidant capacity of compounds. As reviewer`s have said, ORAC is based on HAT (hydrogen atom transference), so if components from acai extract is more directed to SET (single electron transference), antioxidant activity of this fruit would be underestimated. Thus, in this work, authors have decided to measure antioxidant activity of fruit extracts through a method with a unique mechanism, as FRAP that is based on SET mechanism, and through DPPH and TEAC methods that are able to analyze antioxidant activities of extracts through both antioxidant mechanisms of action, SET and HAT. On the other hand, ORAC is well known and, as reviewer have confirmed, it has been already done. ORAC studies from Schauss et al (2006) was mentioned in discussion section and reference included (Ref. 43).
Line 114. What modifications were made? Important to know as any modification of the method if carried out correctly could affect results.
Modifications were included in the manuscript (lines 126,127).
Line 122. What does “richest” mean? Suggest using a scientific term, as this sounds like puffery. Corrected
Line 217. Which version? Version 11.5 was used and sentence in the manuscript was corrected and this information was incorporated.
Line 236/ Throughout the manuscript, the authors should replace “acai-berry extract” with “acai-fruit pulp extract”, as that is the source material for the author’s study. See line 445, where “berries” is used; wherever reference to acai as a berry is found in the submission, replace it with the word “fruit.” Corrected
Line 238-240. Neither calafate or maqui come close to the published ROS scavenging activities reported for acai in the literature. The authors may be relying on the paper by Speisky H, et al, for their opinion. From a commercialization standpoint it is understandable why this misinformation wouldt be perpetuated by promoters of these two food products, given how consumers are unfamiliar with who to interpret ORAC values. However, a careful review of the literature will reveal that lyophilized-acai-pulp has greater ROS scavenging activity. The ORAC for acai is reported; see: Antioxidant capacity and other bioactivities of the freeze–dried Amazonian palm berry, Euterpe oleraceae Mart. (Açai). Journal of Agricultural and Food Chemistry, 2006; 54(22); 8604–8610. The authors are likely relying on the paper by Speisky H, et al, for their opinion. As the authors will see on page 8,607 of the paper in J Agric Food Chem, the combined value for hydrophilic and lipophilic scavenging activity for lypohilized-acai is c. More significantly, the SOD scavenging is reported at 1,614 - the highest reported value for a fruit or berry.
Authors think reviewer has misunderstood the idea proposed in the manuscript. Based on Speisky et al. studies, acai is positioned with highest values regarding to its phenolic content than other fruits as calafate and maqui. Nevertheless, authors have considered the work cited by the reviewer and this reference was included during the discussion of antioxidant activities results (lines 274-277).
Ultimately, what is important is evidence of demonstrated experimental evidence of a food’s beneficial health effects. A current review of the literature on Acai shows significantly more in vivo evidence of its benefits than for maqui, and certainly for calafate, whose production is limited and yet to become a significant exported food for Chile or Argentina.
Authors corrected the manuscript in order to avoid misunderstanding between acai and maqui/calafate. Besides, introduction has included several studies which evidence health benefits of this fruit.
Line 415-416. Suggest the authors include information on the short-comings (limitations) of relying on this method to evaluate bioaccessibility and explain how the term bioaccessibility is different from that of bioavailability.
Authors thank reviewer´s comment. Following his/her recommendation, a new paragraph was included to clarify this issue (lines 437-442). However, authors consider that it is not relevant to incorporate the limitations of the methodology used (static in vitro digestion) because it is widely known their advantages and disadvantages (Bohn et al, 2018; den Abbeele, 2019). Besides, this methodology is widely currently used to carry out studies on the positive and negative effect of food structure, composition and processing on nutrient and bioactive compounds bioaccessibility (Alegria et al, 2015).
In this work, bioaccesibility of phenolic compounds was evaluated since released phenolic content from the food matrix (Açaí-fruit extract or encapsulated) within the gastrointestinal tract was measured, which it will be available for their intestinal absorption. On the other hand, bioavailability is defined as the fraction of ingested component available at the site of action for utilization in normal physiological functions, and it is commonly determined through in vivo assays (Alegría et al., 2015).
. Bohn, T., Carriere, F., Day, L., Deglaire, A., Egger, L., Freitas, D., Golding, M., Le Feunteun, S., Macierzanka, A., Menard, O., Miralles, B., Moscovici, A., Portmann, R., Recio, I., Rémond, D., Santé-Lhoutelier, V., Wooster, T.J., Lesmes, U., Mackie, A.R., Dupont, D. Correlation between in vitro and in vivo data on food digestion. What can we predict with static in vitro digestion models?
(2018) Critical Reviews in Food Science and Nutrition, 58 (13), pp. 2239-2261.
. den Abbeele, P. Can dynamic in vitro digestion systems mimic the physiological reality? (2019) Critical Reviews in Food Science and Nutrition, 59 (10), pp. 1546-1562.
. Alegría, A., Garcia-Llatas, G., Cilla, A. Static digestion models: General introduction (2015) The Impact of Food Bioactives on Health: In Vitro and Ex Vivo Models, pp. 3-12.
Lines 445-454. After reading the submission and re-reading it a second time, one continues to search for what the compelling evidence is of the superiority for encapsulation especially given the lack of comparative data to other methods of preservation.
Authors compared effectiveness of encapsulated acai extract over commercial dehydrated acai and its lyophilized concentrated extract when exposed to baking and sterilization through autoclaving conditions. The purpose of this manuscript was not the comparison between different methods of preservation, but the advantage of encapsulating the extract during these thermal treatments at high temperatures and during digestive phases.
Nevertheless, authors thank reviewer´s comment and the comparison with other methods of preservation will be taken into account in the future.
Authors should carefully review each reference. For example, on line 483, the word “comercial” is misspelled. Also, use of lower and upper case for titles of articles should be consistent and according to author guidelines.
This reference was corrected, and References were reviewed.

Reviewer 3 Report
The manuscript described the enhancing thermal stability and bioaccesibility of
Açaí-berry polyphenols through electro-hydrodynamic encapsulation into zein electrosprayed particles. This work aims to investigate to improve thermal stability and bioaccesibility of Açaí-berry polyphenols. The results have a positive effect on thermal stability and bioaccesibility of Açaí-berry polyphenols. The manuscript provided a series of data regarding the enhancing thermal stability and bioaccesibility of Açaí-berry polyphenols. These results may provide a valuable reference for functional food and nutraceutical applications. Therefore, I suggest that the manuscript may be considerable to publish in antioxidants after major modifications:
Giving the “abbreviations section” in beginning of this manuscript. Page 3, line 91: 1:300 (solid:solvent) ratio….=> giving units in ratio. Page 9, line 315: in Figure 2, meaning (or name) of three curves is not specified. Page 11, line 386: in Figure 4, please explain clearly why phenolic content loss values are the significant difference between sterilization and baking for AÇEXT. Page 12, 3.7 section: if author can provide data in phenolic content of açaí samples from heat treatment after in vitro digestive, the manuscript will be more valuable with the more profits of encapsulation process comparing to un- encapsulation.
Author Response
Regarding to this evaluation, authors have extensively revised introduction, methodology, results and conclusions in order to improve the manuscript.
Comments and Suggestions for Authors
The manuscript described the enhancing thermal stability and bioaccesibility of Açaí-berry polyphenols through electro-hydrodynamic encapsulation into zein electrosprayed particles. This work aims to investigate to improve thermal stability and bioaccesibility of Açaí-berry polyphenols. The results have a positive effect on thermal stability and bioaccesibility of Açaí-berry polyphenols. The manuscript provided a series of data regarding the enhancing thermal stability and bioaccesibility of Açaí-berry polyphenols. These results may provide a valuable reference for functional food and nutraceutical applications. Therefore, I suggest that the manuscript may be considerable to publish in antioxidants after major modifications:
Giving the “abbreviations section” in beginning of this manuscript.
Authors agree with the fact including an “Abbreviation section” will contribute to a better understanding. Thus, this section was included after Supplementary Material and Antioxidants Journal Editors will decide the appropriate place.
Page 3, line 91: 1:300 (solid:solvent) ratio….=> giving units in ratio.
Units were incorporated.
Page 9, line 315: in Figure 2, meaning (or name) of three curves is not specified.
Figure Caption of Figure 2 was corrected.
Page 11, line 386: in Figure 4, please explain clearly why phenolic content loss values are the significant difference between sterilization and baking for AÇEXT.
As TGA analysis show in Figure 3B, AÇEXT presented an early degradation that started at approximately 100 °C with a maximum degradation at 162.5 °C. Thus, its degradation after sterilization because it occurred at 121 °C (before maximum degradation) was low (approx. 10%). Nevertheless, because baking temperature occurred at 180 °C which is higher than degradation temperature, AÇEXT degradation significantly increased to 55% approx.
Authors thank reviewer`s comment and add s brief explanation to clarify this point. Moreover, this comment implied the observation Figure 4 needed to be corrected regarding statistical differences between phenolic content losses after sterilization and baking processes of ZN/ AÇEXT.
Page 12, 3.7 section: if author can provide data in phenolic content of açaí samples from heat treatment after in vitro digestive, the manuscript will be more valuable with the more profits of encapsulation process comparing to un- encapsulation.
Authors appreciate the reviewer's proposal and agree that evaluating the bioaccessibility of samples after thermal processes is very interesting, and will be taken into account for future works. However, in this work the main objective was characterizing through the antioxidant activities and the phenolic content of açaí fruit extracts and subsequently encapsulating it through the electrospinning technique using an edible polymer. The efficiency of this encapsulation was evaluated by subjecting this zein containing extract capsules with its controls (açaí fruit and lyophilized extract) to thermal treatments and in vitro digestive processes, but separately. The sum of both processes may imply very low results and difficulty to quantify, but this idea will be definitely taken into account for future works.

Round 2
Reviewer 2 Report
The authors made significant improvements to support the readability in the revised manuscript submission. It will be interesting to see if encapsulation of the phytochemicals and nutrients in the fruit's pulp will yield equivalent or superior therapeutic outcomes in in vivo experiments and clinical trials.
Reviewer 3 Report
Authors should clearly check the English language and style.